# Bootstrapping Unsupervised Deep Music Separation from Primitive Auditory Grouping Principles

Prem Seetharaman [1]   Gordon Wichern [2]   Jonathan Le Roux [2]   Bryan Pardo [1]

## Abstract

Separating an audio scene, such as a cocktail party with multiple overlapping voices, into meaningful components (e.g., individual voices) is a core task in computer audition, analogous to image segmentation in computer vision. Deep networks are the state-of-the-art approach. They are typically trained on synthetic audio mixtures made from isolated sound source recordings so that ground truth for the separation is known. However, the vast majority of available audio is not isolated. The human brain performs an initial segmentation of the audio scene using primitive cues that are broadly applicable to many kinds of sound sources. We present a method to train a deep source separation model in an unsupervised way by bootstrapping using multiple primitive cues. We apply our method to train a network on a large set of unlabeled music recordings to separate vocals from accompaniment without the need for ground truth isolated sources or artificial training mixtures. A companion notebook with audio examples and code for experiments is available[1][2].

## 1. Introduction

A fundamental problem in computer audition is audio source separation: isolating a sound producing source (e.g., a singer) or group of sources (e.g., a backing band) in an audio scene (e.g., a music recording). Source separation is important for building machines that can perform perceptual audio tasks on realistic audio inputs with human-level performance. Deep models are the current state-of-the-art

[1]Department of Computer Science, Northwestern University, Evanston, IL, USA [2]Mitsubishi Electric Research Labs, Cambridge, MA, USA. Correspondence to: Prem Seetharaman <prem@u.northwestern.edu>.

*Published at the workshop on Self-supervision in Audio and Speech at the $37^{th}$ International Conference on Machine Learning*, Vienna, Austria. Copyright 2020 by the author(s).

[1]https://github.com/pseeth/bootstrapping-computer-audition

[2]This work has made use of the Mystic NSF-funded infrastructure at Illinois Institute of Technology, NSF award CRI-1730689.

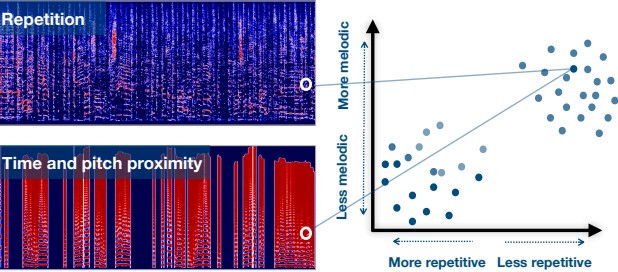

*Figure 1.* The process for making a primitive embedding. A set of primitive algorithms is run on the mixture. Each algorithm produces a mask with values between 0 (blue) and 1 (red) indicating how it is segmenting the auditory scene. Here, we show primitive clustering for two primitives. Together, they map each time-frequency point into a 2D embedding space, shown on the right. The marked point was classified by the primitives as melodic and not repetitive, indicating that it likely belongs to the vocals.

for source separation of mixtures of speech or music (Stöter et al., 2018). These models are trained on thousands of synthetic mixtures of isolated recordings of musical instruments and voices. Using synthetic mixtures guarantees that the ground truth isolated sources are known. However, most sounds in the world do not occur in recording studios and most available audio recordings (e.g., field recordings, YouTube videos) do not have available decompositions into their isolated components.

A system that learns in an unsupervised way, bootstrapping from audio scenes where no pre-separation into isolated sources is available, would be foundational to building systems that can learn from broadly available sources of audio (e.g., audio from YouTube, or live microphone input) containing a much larger range of sounds in a much larger range of mixtures than is possible with synthetic mixtures. This should lead to more robust source separation, able to separate more classes of sounds in more kinds of mixtures.

The way deep source separation models learn is in stark contrast to how humans learn to segregate audio scenes (Bregman, 1994): sources are rarely presented to us in isolation and almost never in "mixture/reference" pairs. There is experimental evidence that the brain uses primitive cues (e.g., direction of arrival, repetition, proximity in pitch and

time) that are independent of the characteristics of any particular sound source to perform an initial segmentation of the audio scene (McDermott et al., 2011a). The brain could use such cues to separate at least some scenes to some extent, and use that information to train itself to separate more difficult scenes (McDermott et al., 2011b).

Source separation algorithms based on primitives are well-studied (Wang & Brown, 2006). These include algorithms based on common-fate (Seetharaman et al., 2017), melodic contours (Salamon & Gómez, 2012), direction of arrival (Fitzgerald et al., 2016), repetition (Liutkus et al., 2012; Rafii & Pardo, 2013) and harmonic/percussive timbre (Fitzgerald, 2010).

Le Roux et al. (2013) proposed to use ensemble learning for speech enhancement, combining time-frequency masks corresponding to outputs of multiple enhancement algorithms using a shallow classifier. Liutkus et al. (2014) put primitive separation algorithms into a framework where elements of the time-frequency representation are clustered using a set of primitive-specific proximity kernels. Their approach assumes each source can be separated using a single primitive and does not provide a way of estimating confidence in that primitive. Other approaches (Bach & Jordan, 2006; Manilow et al., 2017; McVicar et al., 2016) learn how to combine primitives but require ground truth to do so.

We build on deep learning source separation models (Luo et al., 2017; 2020) that use time-frequency embeddings to separate sources (Hershey et al., 2016; Luo et al., 2018). Training these models typically requires ground truth. Previous efforts (Drude et al., 2019; Seetharaman et al., 2019; Tzinis et al., 2019) to train source separation models without ground truth all learn models to separate speech in stereo mixtures using only the direction-of-arrival primitive to generate training data. These approaches fail when direction-of-arrival is unreliable (e.g., highly reverberant audio) or unavailable (monophonic audio).

**Overview:** Our approach has three components: the labeller, the student, and the teacher. We first develop a clustering-based labeller that allows combining multiple perceptually-inspired primitive cues, which separates better than any primitive by itself. We then propose a confidence measure that estimates the effectiveness of any clustering-based source separation approach, without ground truth. Finally, a teacher uses the confidence measure to select outputs from the primitive clustering system to recombine into training examples. This curriculum is used to train the student (a deep model) which outperforms the primitive clustering labeller. We then reuse this framework to bootstrap a second student using teacher-selected separation estimates from the first student as training data, outperforming the first student. Although we focus our presentation on music mixtures, our approach is applicable to other domains.

## 2. The Labeller

The first component of our system is the labeller, which separates mixtures using an ensemble of primitive auditory cues in a method we call *primitive clustering* (Fig. 1). We apply the labeller to a magnitude spectrogram, a widely-used representation that indicates the magnitude of the audio at each point in time and frequency. Given a music mixture $X$, each primitive-based algorithm produces a soft time-frequency mask that assigns to each time-frequency (TF) point $(t, f)$ in the mixture a real number ranging from 0 (if it is dominated by the accompaniment) to 1 (if it is dominated by vocals). The masks produced by the primitives are used to create an embedding of each TF point in the mixture, where each dimension of the embedding contains the soft mask value at that TF point according to one of the primitives. Given $D$ primitives, each TF point $X(t, f)$ is thus mapped to a $D$ dimensional vector, which we denote as $\mathcal{F}(X)(t, f)$.

To turn the multi-dimensional embedding representation into a single mask that takes into account all the primitives, we choose to use a clustering-based approach, specifically so we can not only cluster points into accompaniment- and vocals-dominated regions but also predict the separation quality. We use here an approach related to soft K-Means clustering (Jain, 2010). However, since our goal is not to find cluster centroids as in K-means, but rather calculate a score for each embedding vector based on consensus of the different primitives, we fix the means of the two clusters to $\mu_0 = [0]^D$ and $\mu_1 = [1]^D$. These points are where the primitive separation algorithms all strongly agree on how to assign a TF point to either the accompaniment (0) or vocals (1). We then use the distance of the primitive embedding $\mathcal{F}(X)(t, f)$ of every TF point $(t, f)$ to $\mu_0$ and $\mu_1$ to calculate the soft mask for each source $M_k(t, f)$ as

$$M_k(t, f) = \frac{e^{-\beta \mathcal{D}(\mathcal{F}(X)(t,f), \mu_k)}}{\sum_j e^{-\beta \mathcal{D}(\mathcal{F}(X)(t,f), \mu_j)}} \tag{1}$$

where $\mathcal{D}(x, y)$ is the Euclidean distance between points $x$ and $y$. This maps distances in the embedding space to values between 0 and 1. $\beta$ ($= 5.0$, here) controls the hardness of the decisions made by the clustering algorithm.

## 3. The Teacher

Our goal is to train a deep learning model to separate audio mixtures without ground truth. One simple way to do this would be to train the network directly from the TF labels (i.e., soft mask values) the labeller generates for every mixture in a large pool of unlabeled examples. If the labeller fails on some of these mixtures, the network will be trained with bad data. To mitigate this problem, the teacher uses two strategies: a confidence measure used to identify failure cases and exclude them from training, and a data augmenta-

tion process to create high-quality training examples from the separated outputs produced by the labeller.

## 3.1. Confidence measure

Our deep models and all the primitive source separation methods in this paper use clustering-based source separation. Time-frequency points are mapped to an embedding space. In a good embedding, clusters are distinct, and points in the same cluster correspond to the same sound source. By analyzing the embedding space, we can estimate confidence in the performance of any clustering-based source separation algorithm without the need to compare to ground truth sources. The confidence measure $c(X)$ over the set of embedded TF points in an audio example $X$ has two parts: the silhouette score and posterior strength, which are multiplied together: $c(X) = s(X)P(X)$.

### 3.1.1. SILHOUETTE SCORE

The silhouette score (Rousseeuw, 1987) produces a score for every point in a dataset that corresponds to how well that point is clustered. To compute the silhouette score, let us first assume we have a partition of dataset $X = \bigcup_{k=1}^{K} C_k$ into $K$ clusters. For a data point $x_i$ in cluster $C_k$, we compute the following terms:

$$a(x_i) = \frac{1}{|C_k| - 1} \sum_{x_j \in C_k, i \neq j} d(x_i, x_j),$$

$$b(x_i) = \min_{o \neq k} \frac{1}{|C_o|} \sum_{x_j \in C_o} d(x_i, x_j).$$

$a(x)$ is the mean distance (using a distance function $d$) between $x_i$ and all other points in $C_k$, and $b(x)$ is the mean distance between $x_i$ and all the points in the nearest neighbor cluster. The silhouette score $s(x)$ ranges from $-1$ to $1$ and is defined as:

$$s(x) = \frac{b(x_i) - a(x_i)}{\max(a(x_i), b(x_i))} \text{ if } |C_k| > 1, \qquad (2)$$

and $s(x_i) = 0$ if $|C_k| = 1$. While the soft assignment from Eq. (1) is used when separating, in order to apply the silhouette score to primitive clustering, we assign each point to the cluster (vocals or accompaniment) whose center it is closest to.

Computing the silhouette score for all points in a typical auditory scene (millions of points) is intractable. Instead, we compute the silhouette score on a sample of $Q$ points and take the mean as the score for the overall example. With a relatively small $Q$, we can approximate the mean silhouette score efficiently. We set $Q = 1000$, and sample these $Q$ points from the loudest 1% of time-frequency bins in the mixture spectrogram, to focus our estimate on elements that are perceptually prominent.

### 3.1.2. POSTERIOR STRENGTH

For every point $x_i$ in set $X$, the clustering algorithm produces soft assignments $\gamma_{ik} \in [0, 1]$ that indicate the membership of the point $x_i$ in each cluster $C_k$. $\gamma_{ik}$ is also called the *posterior* of the point $x_i$ in regards to cluster $C_k$. The closer $\gamma_{ik}$ is to 0 (belongs to a different cluster) or 1 (belongs to this cluster), the more confident the assignment of that point. For a point $x_i$ with corresponding $\gamma_{ik}$ for $k \in [0, 1, ..., K]$, we compute

$$P(x_i) = \frac{K(\max_{k \in [0,...,K]} \gamma_{ik}) - 1}{K - 1}, \qquad (3)$$

where $K$ is the number of clusters, and $P(x_i)$ is the *posterior strength*, as it captures how strongly a point is assigned to any of the $K$ clusters. This equation maps points that have a maximum posterior of $\frac{1}{K}$ (equal assignment to all clusters) to confidence 0, and points that have a maximum posterior of 1 to confidence 1. To compute a single $P(X)$ score over $X$, we take the mean posterior strength across the top 1% of points, by loudness.

## 3.2. Constructing the curriculum

For each mixture, we apply primitive clustering to obtain each separation estimate (isolated voice or background music) from each unlabeled audio mixture. A threshold on the confidence for each estimate is used to determine whether to use it in training. To create training examples, we take vocals and accompaniment estimates and remix them to create new mixtures. We remix in two ways: *coherent* and *incoherent*. Coherent mixing occurs when the vocals and accompaniment estimates come from the same song at the same time instant (e.g., it sounds like conventional music). Incoherent mixing mixes estimates obtained from different songs. A single training example then consists of the created mixture as the input and the primitive separation estimates as the target.

We use Scaper (Salamon et al., 2017) for data augmentation. Each training mixture is 10 s long and at 16 kHz sample rate, with a signal-to-noise ratio between vocals and accompaniment chosen randomly between 0 and 10 dB. Estimates are pitch shifted randomly between $-2$ and $+2$ semitones and time stretched randomly between 0.8x and 1.2x the original length. In the coherent mixing case, both estimates are pitch shifted and time stretched by the same amount.

## 4. The Student

The student, following a Chimera++ architecture (Wang et al., 2018), is a deep neural network consisting of multiple recurrent layers and two output heads. The first output head is a mask-inference head. This head is used for performing the actual source separation, by masking out TF points

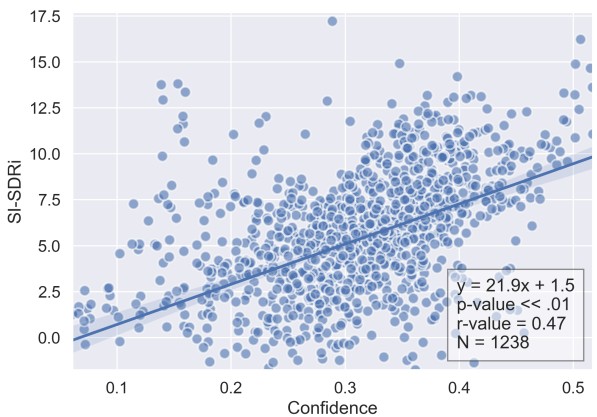

*Figure 2.* Relationship between confidence measure and actual performance for music mixtures, using primitive clustering to separate each mixture. Each of the 1238 points represents the vocals produced by primitive clustering for one 30-second excerpt from the MUSDB training set. The blue line is the line of best fit.

in the spectrogram that do not correspond to the desired source. It is trained by comparing an estimated spectrogram for a separated source $\hat{S}$ with the "ground truth" source $S$ obtained from the unsupervised labeller:

$$\mathcal{L}_{\text{MI}} = \frac{1}{N} \Big\| |\hat{S}| - |S| \Big\|_1, \tag{4}$$

where $N$ is the number of TF points in $S$.

The deep clustering head is trained via the whitened K-Means objective (Wang et al., 2018), which compares the affinity matrix of the embeddings $V$ for all TF points with that of the "ground truth" binary assignments $Y$:

$$\mathcal{L}_{\text{WKM}} = D - \text{tr}((V^T V)^{-1} V^T Y (Y^T Y)^{-1} Y^T V) \tag{5}$$

where $D$ is the embedding size. The two objective functions are linearly combined, with a weighting factor $\alpha$:

$$\mathcal{L}_{\text{MI+WKM}} = \alpha \mathcal{L}_{\text{MI}} + (1 - \alpha) \mathcal{L}_{\text{WKM}}. \tag{6}$$

In this work, $\alpha$ is chosen to be 0.75. We use the dual path recurrent layer (DPRNN) proposed by Luo et al. (2020). We use a 3 layer DPRNN, with 300 hidden units. The input to the network is the magnitude STFT with window size 512, hop length 128, and the square root of the Hann window as the window function. We apply an instance normalization layer on the input STFT so the network is resilient to loudness changes. Before the recurrent stack, the linear-spaced frequency STFT is projected to 300 mel-spaced frequencies. The output of the deep clustering head consists in 20-dimensional embeddings of the TF points and is used to estimate confidence in the separation produced by the network in the same way confidence is estimated for primitive clustering.

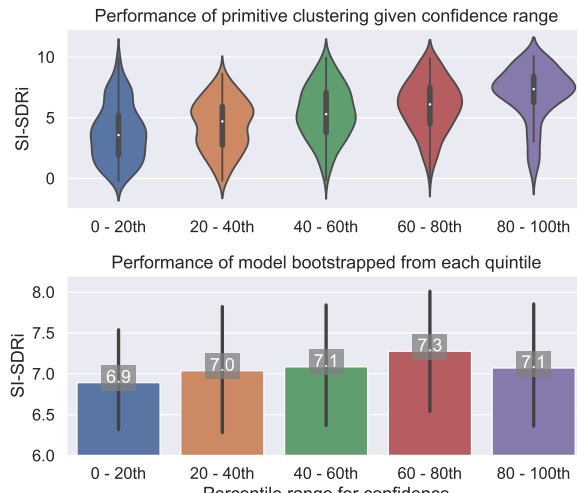

*Figure 3.* In the top plot, we take all the estimates produced by primitive clustering on the MUSDB train set and split them into quintiles according to the confidence range. Higher quintiles according to confidence also have higher SI-SDRi distributions. In the bottom plot, we show the effect of bootstrapping a model from curriculum created from each quintile in the top plot. The network bootstrapped from the lowest quintile has poor performance, indicating that excluding low-confidence sources is helpful.

## 5. Experiments

Our experiments are designed to answer whether primitive clustering outperforms any one primitive by itself, whether the confidence measure is a good unsupervised estimator of separation quality, and whether the student deep networks outperform the primitive labeller.

### 5.1. Data and hyperparameters

In our experiments, we use a dataset with ground truth separated sources so we can benchmark the effectiveness of the unsupervised training. We use the 150 song MUSDB dataset (Stöter et al., 2018), split into training (86 songs), validation (14 songs), and test (50 songs) sets. We augment this dataset with songs scraped from YouTube ( 800 songs), for which we do not have ground truth separated sources. Primitive clustering uses four primitives: micromodulation and repetition (Seetharaman et al., 2017), time/pitch proximity (Salamon & Gómez, 2012), and harmonic/percussive timbre (Fitzgerald, 2010). Vocal separation performance is measured using scale-invariant source-to-distortion ratio *improvement* (Le Roux et al., 2019) (SI-SDRi) over using the mixture as the estimate. All separation methods were implemented using the *nussl* toolkit (Manilow et al., 2018) and are evaluated on the MUSDB test set. Each deep network was trained for 20k iterations, with a batch size of 20, and the Adam (Kingma & Ba, 2014) optimizer with learning rate 1e-3.

| Notes | Training data | SI-SDRi |
|---|---|---|
| **Primitives** | | |
| Timbre | n/a | −0.7 |
| Proximity | n/a | 3.4 |
| Micromodulation | n/a | 6.2 |
| Repetition | n/a | 6.4 |
| Prim. clustering | n/a | **6.8** |
| **Bootstrapping from primitive clustering** | | |
| Bootstrap | Q4 of MUSDB | 7.3 |
| Bootstrap | MUSDB + YT (all) | 7.5 |
| Bootstrap | MUSDB + YT (Q2-Q5) | **7.6** |
| **Bootstrapping from bootstrapping** | | |
| Bootstrap | MUSDB + YT (Q2-Q5) | **7.7** |
| **Training with ground truth** | | |
| From scratch | MUSDB VAL (800 iters) | 3.0 |
| Bootstrap init. | MUSDB VAL (800 iters) | **8.5** |
| From scratch | MUSDB (20k iters) | 10.0 |
| Bootstrap init. | MUSDB (20k iters) | **10.4** |

*Table 1.* Vocal separation performance of all approaches in terms of SI-SDR improvement [dB] on the MUSDB test set. In the table, MUSDB is the MUSDB training set, MUSDB VAL is the MUSDB validation set (which has only 14 songs), and YT is the dataset of songs scraped from YouTube. "Q$n$" ($n = 2, \ldots, 5$) refers to the $n$-th quintile by confidence. For SI-SDRi, higher values are better.

## 5.2. Results

In Table 1, we show the performance of each primitive on the MUSDB test set. We find that the best-performing primitive is repetition. The combination of all 4 primitives ("Prim. clustering") is 0.4 dB better than repetition alone.

We now test whether the confidence measure is predictive of separation quality. We split each track in the MUSDB training set into 30 second mixture segments and performed primitive clustering on each segment. For each segment, we applied our confidence measure and computed true separation performance with SI-SDRi. Fig. 2 and 3 show a strong relationship between confidence and SI-SDRi.

Next, we consider whether using the confidence measure to create a curriculum has an impact on the performance of a bootstrapped model. In Fig. 3, we show the performance of a model bootstrapped from curriculum created from estimates with varying confidence. We see that the higher the confidence we have in the estimates used to train a model, the better the performance of the model, to a point. The last quintile has a dip in performance, which we hypothesize is due to the lower diversity of songs in that subset.

In Table 1, we show the results of other training conditions. We find that including additional unlabeled data from YouTube improves performance by 0.2 dB. Using the confidence measure to exclude the bottom quintile from the

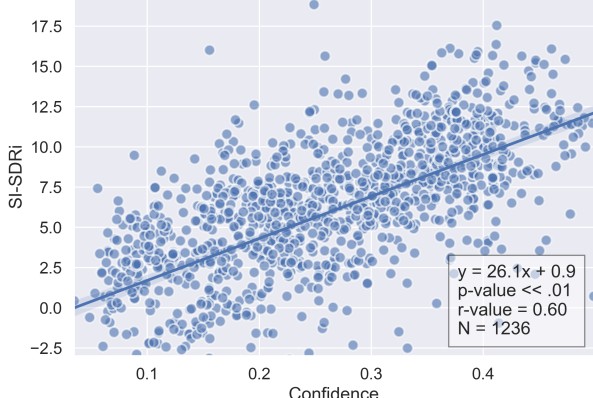

*Figure 4.* Relationship between confidence measure and actual performance for music mixtures, using the bootstrapped deep model to separate each mixture. The confidence measure is calculated using the deep clustering head's output, but the separation estimate is produced by the separation head.

curriculum further improves performance by 0.1 dB. All bootstrapped models outperform the primitive clustering labeller. The separation quality of a deep model can be predicted by applying our confidence measure to the deep clustering output head, as is illustrated by the strong positive correlation in Fig. 4. This lets us train a second deep model, using the first deep model as the labeller. The second trained model results in even further improvement to 7.7 dB, 0.9 dB better than the primitive clustering labeller.

We finally experiment with using the best bootstrapped model as an initialization for training with ground truth isolated sources. We compare these to models that are trained from scratch in Table 1. With very little data – the MUSDB validation set of 14 songs – trained for only 800 iterations, bootstrap initialization leads to a boost in performance to 8.5 dB, 5.5 dB higher than a model trained from scratch for 800 iterations. This continues to hold for more iterations more data, though the gap decreases to 0.4 dB.

## 6. Conclusion

We have presented a method for learning to separate sounds directly from mixtures without ground truth via primitive auditory grouping principles. First, a labeller is applied to a large set of mixtures for which we do not have ground truth. Then, a teacher uses the confidence measure to construct a curriculum. Finally, a student learns from the curriculum. The student network can outperform the primitive labeller by a significant margin. The process is then repeated using the bootstrapped model as the labeller, as the confidence measure can be used for any separation algorithm that is based on clustering, resulting in further improvement. This opens the door to deep separation models that can continuously learn in the wild via bootstrapping.

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
