# OpenReview forum: "Bootstrapping Unsupervised Deep Music Separation from Primitive Auditory Grouping Principles"
_ICML.cc/2020/Workshop/SAS — SAS 2020_

### Official Review · AnonReviewer1 · 2020-06-27
**Unsupervised Deep Music Separation**

**Rating:** 8
**Confidence:** 3

**Review:**

The paper proposes a solution to unsupervised music separation. The solution is based on three stages called labeler, teacher, and student.
The labeler runs N signal processing algorithms that provide some soft masks as an initial solution (each TF points is then represented by a vector of N elements). The TF points are clustered using a standard clustering algorithm to finally generate a single TF mask. The teacher analyzes the clustering solution and provides a confidence measure useful to both discard bad training points and to perform data augmentation. Finally, the student is trained on the resulting TF masks using a dual-path RNN.

The paper is well written and the results are quite solid. The topic of the paper is for sure in line with the topics addressed in the workshop.

Minor things:
- The proposed technique is a combination of three independent stages, where the labeler involves an external algorithm based on signal processing.  As future work, it could be interesting to study a fully neural approach end-to-end trainable.
- The paper mentioned that  "our approach is applicable to other domains".  I think the authors are mainly thinking about speech separation.  It could be great if the author specifies better which domains they are referring to and how the extension could be done

---

### Official Review · AnonReviewer2 · 2020-06-30
**Novel idea with minor details missing**

**Confidence:** 3
**Rating:** 7

**Review:**

The paper introduces a novel unsupervised method for music separation, which consists of a labeler, a teacher, and a student. By leveraging bootstrapping, the model performance on the MUSDB test set is promising.

Pros:
 - The idea is novel, which encourages the usage of bootstrapping and unsupervised training for sound separation tasks.
 - The results are promising, which showcase the importance of confidence measure and bootstrapping.

Cons:
 - Lack of a solid baseline. Although the results look promising, the authors didn't compare their approach with any existing solid baseline.
 - What's  "ground truth" binary assignments Y in formula (5)?
 - Timbre primitive doesn't seem to to be helpful, did the authors try to exclude it from prim. clustering?

The confidence measure would drop the underconfident samples, however, those samples could be considered as "hard negatives" which may boost the performance later on. It would be interesting to leverage those samples in future work.

---

### Official Review · AnonReviewer3 · 2020-07-01
**A single data point that is hard to judge**

**Confidence:** 3
**Rating:** 5

**Review:**

The paper presents a method to bootstrap unsupervised sound source separation by performing clustering on audio data that contains multiple sound sources. Unsupervised sound source separation is a challenging topic, but this paper is not the first one if its kind, even though the particular technical approach seems to be novel and is perceptually motivated. The paper is overall well written.

That said, it is difficult to judge the contribution of this particular approach, because the paper shows that "bootstrapping" the model leads to improvements of 0.9dB to 5.5dB (if I interpret the experiments correctly) over a simple baseline in a very-low data and very-low training regime, while improvements diminish to less than 0.4dB in the case of more fully trained models. The authors also train their models on a mixture of "clean" standard data as well as Youtube data, which seems to make some difference (0.1dB) for the proposed approach, but it is not clear how much the "noisy" data contributes in all the scenarios, and how the contribution scales to bigger training data sets.

To be accessible to a wider audience, the paper should include a clear comparison of the proposed approach to other published work, and show how the proposed work performs on "standard" (if that is possible) setups (training/ testing data and metrics) that have been used in other work. "Music Source Separation in the Waveform Domain", Defossez et al 2019, e.g. also uses 800 Youtube videos - are they the same, or not?

In its present form, the paper is interesting but sort of an isolated data point, that is hard to judge (unless one has been intimately following exactly that line of work for the last years).

---

### Decision · Program_Chairs · 2020-07-01

**Decision:**

Accept

**Comment:**

Dear author(s),

Thank you very much for your submission at the ICML2020@SaS workshop (https://icml-sas.gitlab.io/). Based on the scores assigned by the reviewers, we are happy to notify you that your paper was accepted for the workshop.

Please, address the comments of the reviewers and submit the camera-ready version by July 8. We ask the authors to record a 15min video for your talk. At the workshop, we will have the pre-recorded video as well as a live QA session. It is important to keep this time limit, otherwise, your talk will be automatically cut. The deadline for uploading the video is July 8. The detailed instructions for uploading will follow.

Feel free to contact us for any questions!

Best,

The ICML20@SaS organizers:
Mirco Ravanelli
Titouan Parcollet
Dmitriy Serdyuk
Devon Hjelm
Bhuvana Ramabhadran